# Public health risk communication through the lens of a quarantined community: Insights from a coronavirus hotspot in Germany

**Annika Licht**[1], **Wibke Wetzker**[1], **Juliane Scholz**[1], **André Scherag**[2], **Sebastian Weis**[1,3,4], **Mathias W. Pletz**[3], **Michael Bauer**[1], **Petra Dickmann**[1]*, **the CoNAN study group**[¶]

**1** Department of Anaesthesiology and Intensive Care Medicine, Jena University Hospital, Friedrich Schiller University, Jena, Germany, **2** Institute of Medical Statistics, Computer and Data Sciences, Jena University Hospital, Friedrich Schiller University, Jena, Germany, **3** Institute for Infectious Diseases and Infection Control, Jena University Hospital, Friedrich Schiller University, Jena, Germany, **4** Leibniz Institute for Infection Biology and Natural Product Research, Leibniz Institute for Natural Product Research and Infection Biology, Hans Knoell Institute, Jena, Germany

¶ The membership of the CoNAN study group is provided in the Acknowledgments. The lead authors of the CoNAN study group are S Weis[3,4] (sebastian.weis@med.uni-jena.de) and MW Pletz[3] (mathias.pletz@med. uni-jena.de).

* petra.dickmann@med.uni-jena.de

**Data Availability Statement:** All relevant data are within the paper and its Supporting Information files. The data were collected in its original

## Abstract

### Background

Quarantine is one of the most effective interventions to contain an infectious disease outbreak, yet it is one of the most disruptive. We investigated the quarantine of an entire village to better understand risk communication requirements for groups.

### Methods

We conducted a cross-sectional, mixed-methods survey study on a single cohort of adult residents in Neustadt am Rennsteig, Germany, six weeks after the removal of a 14-day mandatory community quarantine. The survey response rate was 33% (289/883 residents).

### Findings

Survey participants reported a lack of information on the quarantine implementation process. What authorities communicated was not necessarily what residents desired to know. While inhabitants used social media and telephones to communicate with each other, the official information sources were regional radio, television, newspapers and official websites. Public health authorities did not employ social media communication to engage with their communities. Despite a lack of information, the majority of respondents stated that they had complied with the quarantine and they expressed little sympathy for those who violated the quarantine. After lifting the quarantine, many respondents continued to avoid places where they suspected a significant risk of infection, such as family and friends' homes, doctor's offices and grocery stores.

language German and is available in a translated version.

**Funding:** This research is part of a research group that was granted funding by the Free State of Thuringia: #5575/2-1 63952/2020 and #5526/32-4-2. The funders had no role in study design, data collection and analysis, decision to publish, or preparation of the manuscript.

**Competing interests:** The authors have declared that no competing interests exist.

## Interpretation

The survey participants utilised existing social networks to disseminate vital information and stabilise its group identity and behaviour (quarantine compliance). The authorities communicated sparsely in a unidirectional, top-down manner, without engaging the community. Despite the lack of official information, the social coherency of the group contributed to considerate and compliant conduct, but participants expressed dissatisfaction with official leadership and asked for more attention.

## Conclusion

Public health risk communication must engage with communities more effectively. This necessitates a deeper comprehension of groups, their modes of communication and their social needs.

## Introduction

In December 2019, a new coronavirus emerged—severe acute respiratory syndrome coronavirus 2 (SARS-CoV2)—which causes the disease COVID-19. The virus spread rapidly and was declared a public health emergency of international concern (PHEIC) on 30/01/2020 by the Director-General of the World Health Organization (WHO) [1]. At the beginning of the pandemic, the routes of viral transmission were unknown [2]. Thus, different possibilities, such as droplet infection, contact and airborne transmission, were discussed, and general contact precautions were recommended to reduce transmission [3].

Physical distancing ('social distancing') is recommended as an effective, non-pharmaceutical intervention to reduce the spread of pathogens in the absence of more specific infection control interventions [4]. Isolating ill and infectious people is an established public health and medical strategy to protect healthy communities from contracting diseases; quarantine is a precautious intervention, intended to separate a potentially infectious person from their surroundings [5]. Both strategies are effective infection control interventions that contribute to reducing the spread of pathogens [6]. However, both strategies require risk communication to instruct contagious (isolated) or potentially contagious (quarantined) individuals. These health-related communications between health professionals and individuals are subject of a large body of risk communication research [7–10].

In contrast to individual health consultations, interventions during public health emergencies may require different risk communication styles. This is especially true for groups or entire villages, which may be regarded as an intermediate level between individual communication and general public communication. Communities—and rural remote villages in particular—often have closely tied communication and support networks [11]. Neighbourhoods, circles of friends and colleagues and communities are well-established networks in which information, rumours and opinions circulate rapidly. The significance and accessibility of social media exacerbate this situation: Facebook status updates and Twitter messages are faster and more widely disseminated than news and reports based on conventional research and validation mechanisms [12].

Regarding infectious diseases, this social connectedness (or social coherence) is a double-edged sword: it can promote disease transmission because people spend time together and the perception may be that people you see frequently and are acquainted with do not pose any

danger ('friend-shield effect') [13]. Closely knit communities have been extraordinarily affected by the current pandemic, resulting in major hotspots. The social fabric of communities, however, offers swift, information-sharing networks and facilitates communication, thus allowing for effective infection control interventions [14]. The role of social media has been negatively framed as contributing to misinformation and causing an infodemic [15, 16]. Yet, the beneficial effect of this fast, low-level communication tool has not been fully leveraged for public health risk communication purposes [17, 18].

Tapping into and investigating these community and communication structures could generate new insights into how to better engage communities during public health emergencies and inform public health risk communication [19].

Public health interventions are only successful when the public or particular groups cooperate and comply [9–11, 20]. Yet, the risk perception of laypeople and public health professionals may differ; effective risk communication is pivotal in achieving infection control. Risk communication has a unique function in the field of public health and is one of the core capacities within the International Health Regulations (IHR)—the internationally binding legislative framework for the management of health emergencies [21].

The concept and strategies of risk communication have undergone relevant changes in recent years, from top-down command-and-control communications to community-centred participatory approaches [22]. In the aftermath of the last major PHEIC (i.e. the Ebola outbreak in West Africa), WHO spearheaded a conceptual redesign of risk communication and community engagement (RCCE) as a response to the need for more successful risk communication. According to WHO guidance, risk communication is understood as a governance approach built across the three technical axes of information, communication and coordination [23]. This modern risk communication approach aims to be more adaptable to the respective populations and their needs, taking into account its responsibility for successful public health management [24]. The major shift in WHO's redesign was the framing of risk communication [7, 25]. While previous recommendations relied on basic communication techniques, such as speaking loudly and clearly with unambiguous messages through established channels [26], the modern approach relies on a jointly owned narrative that frames risk communication in a way that is relevant to and meaningful for communities [27]. Building on this framing, risk communication can then be designed to best meet media reception and utilise the right language. The overall aims of risk communication are to contribute to building trust and interacting with community leaders and local infrastructures, to integrate risk communication into local organisations (e.g. by adapting measures to local needs) and to create structures that allow for mutual exchange of information between local stakeholders and public health officials [28]. WHO, among others, integrated the theoretical concept of RCCE into their guidance [29].

With the advent of social media, communication among equals—without a hierarchical gradient (expert-public)—has become more prevalent [15]. This is accompanied by a high degree of authenticity, as everyone can swiftly and directly report on their experiences [28]. However, this may also have a major equalising effect on traditional authorities and institutions [17]. A press release from an esteemed public health institution is merely a Twitter message competing with other tweets for attention and, above all, speed.

Especially in crisis situations, when there is little verified knowledge and much unverified information circulating anyway, social media acquires a unique power: since everyone can communicate rapidly, the expert authority competes with initial information or rumours that spread quickly [16].

The WHO drew attention to the phenomenon of parallel pandemics: an information pandemic ('infodemic') consisting of an uncontrolled mixture of scientific information, rumours,

misinformation and disinformation circulating on social media that has the potential to confuse the public and undermine infection control behaviour [15–18].

WHO has developed a training course focusing on community engagement and risk communication (https://openwho.org/courses/empowering-communities) that aims to disseminate this novel strategy to public health professionals worldwide.

Given the foregoing theoretical framework [23], we wondered how risk communication during the current pandemic has been implemented and integrated into an overall infection control strategy. We chose a case of community quarantine of an entire village to better understand the perspectives of groups and their requirements and recommendations for public health risk communication. While others highlighted the psychological impact of quarantine [30–32], we were particularly interested in public health risk communication and the role of social media during infodemic public health emergencies. Quarantined hotspots provide a unique, social laboratory-like opportunity for research. Particularly intriguing was the manner in which a remote area with a predominately elderly population, a health department and other institutions utilised social media to implement these public health measures and maintain contact with the quarantined group, as research has highlighted the importance of social media regarding emotional responses to crises [33, 34].

This investigation aims to better understand the implementation of public health risk communication through the lens of an affected community. It further aims to inform public health risk communication guidelines for public health emergencies.

## The case

Neustadt am Rennsteig is a small village with 883 inhabitants (date of survey 05/13/2020–05/16/2020) in a rural area in the German Free State of Thuringia—one of the first major hotspots in Germany for the early phase of the SARS-CoV-2 pandemic. The village is a typical rural village with an elderly population, yet it was considered a high-risk community due to its age groups (47% of the participants are over the age of 60 and 22% of those are 70 years and older) [35].

As the number of cases (six infections) quickly exceeded the capacity of the local public health authority to track, trace and isolate cases, quarantine for the entire village was implemented without warning on a Sunday evening (22/03/2020) using police loudspeaker announcements. In a follow-up flyer distributed on the following Monday (23/03/2020), the announcement of the community quarantine was provided in written form, with a contact number and the duration of the quarantine. Quarantine was declared for all residents of the village of Neustadt am Rennsteig for 14 days beginning on 22/03/2020. During the quarantine period, the number of cases increased to 47 people, three of whom died [14].

## Methods

### Study design and sample selection

We conducted a cross-sectional survey of a single cohort of respondents using both descriptive and exploratory methods. The population of interest was a community under quarantine. Eligibility was determined by the criteria of age (older than 18 years) and place of residence (Neustadt am Rennsteig).

Of the 883 adult residents, 626 participated in the CoNAN study, an epidemiological longitudinal seroprevalence study that was jointly undertaken with our risk communication study [36].

A printed questionnaire was distributed to the participants during a personal briefing, with a request to complete the questionnaire anonymously (see S1 File). A total of 295

questionnaires were returned, of which six had to be excluded because of non-completion (defined as less than 50% of the questions answered). This left 289 valid questionnaires, which thus met the criteria of 278 questionnaires needed for a representative sample for the village (confidence level 95%, margin of error 5 and population size 1,000). Some of the participants did not answer every question (but they did answer more than 50% of the questions), so the number of answers varied. Almost half (47%) of the respondents were aged 60 years or older.

## Measurements

Data were collected using the printed questionnaire shown in the S1 File, which was distributed in combination with the epidemiological questionnaire. The questionnaire contained 16 questions subdivided into four parts. In the first part, the questionnaire asked about demographic details (age, gender and number of people living in the household). The second part of the survey focused on the topic of information. Participants were asked about the use of media before and during quarantine, topics of information from local officials, level of information and specific concerns, such as health (mental), political and economic stability and job security. The third section enquired about communication, specifically interactions with the local authorities. Finally, the questionnaire asked about coordination: acceptance of quarantine as an infection control intervention, compliance with this intervention or views on non-compliance with the quarantine, avoidance of places after the quarantine and respondents' suggestions for similar situations in the future.

The survey analysis was split into quantitative(14) and qualitative analyses. This manuscript presents the qualitative analysis and focusses on the explorative subset of six open-ended questions from the survey. The aim was to better understand preferences regarding information and communication in the community, changes in risky behaviour, concerns and unmet risk communication needs during the quarantine (Fig 1).

**Data analysis.** All free-text responses were analysed using the systematic, rule-guided approach (Mayring) of qualitative text analysis partially quantified for additional orientation (see supplementary material, Table 1) [37]. In one case, the answers to a question about recommendations were correlated with age using linear correlation. Therefore, the handwritten answers were transcribed and imported into Microsoft Excel®.

**Ethics approval.** The study was approved by the ethics committees of Jena University Hospital, Friedrich Schiller University, the respective data protection commissioner (approval number 2020–1776) and the ethics committee of the Thuringian chamber of physicians. The study is registered on the German Clinical Trials Register: DRKS00022416. Informed consent was provided in writing; only adults (aged 18 or older) were included in the study.

## Results

### Information

Overall, 27% (22/80) respondents felt well informed; of the 73% with further information needs, 26% (21/80) did not specify what the additional information need was; 16% (13/80) required more information on the quarantine implementation process, 9% (7/80) required more epidemiological information, 9% (7/80) required more information on supply/daily life/support, 7% (6/80) wanted more information on test facilities and 5% (4/80) wanted more information on appropriate behaviour (Fig 2).

Asked to provide suggestions for information topics (Table 1), 51 participants responded that they required more general information, while 20 indicated that it would be helpful to have more information about the quarantine schedule. Four statements reflected participants' concerns about fake news or sensational journalism.

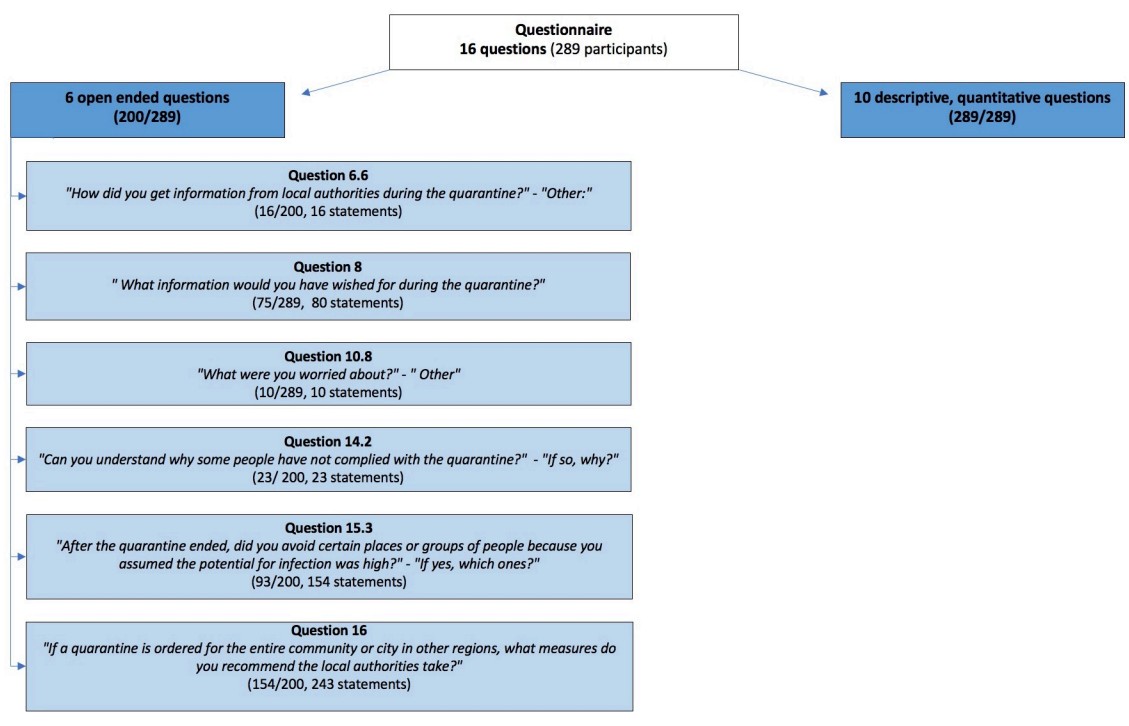

**Fig 1. Overview of qualitative analysis.**

## Communication

Regarding communication, participants highlighted communication modalities, such as the timing and communication formats of the quarantine announcement, including loudspeaker announcements rather than personal communication.

Respondents recommended different modalities for implementation of the quarantine (14/29) and indicated that it should have been announced earlier than it was (8/29). The participants also stressed the importance of the use of different communication styles and formats, such as digital, personal and print communications (Table 2).

## Coordination

**Compliance.** Investigating the community's opinions about how people complied with quarantine, 90% (255/282) of participants said they could not understand why people did not comply with quarantine. Explanations for non-compliance were categorised into intrinsic (referring to individual characteristics) and extrinsic (referring to external reasons) factors and ranged from an individual's inability to understand the concept of quarantine to different prioritisation of activities (e.g. attending social gatherings or responding to family emergencies) (Table 3).

**Table 1. Recommended topics to communicate during quarantine.**

| Category | Total number of statements | Subcategory | Number of statements |
|---|---|---|---|
| **Information** | 75 | Quarantine-schedule | 20 |
| | | Fake news or sensational journalism | 4 |
| | | More information (in general, not specified) | 51 |

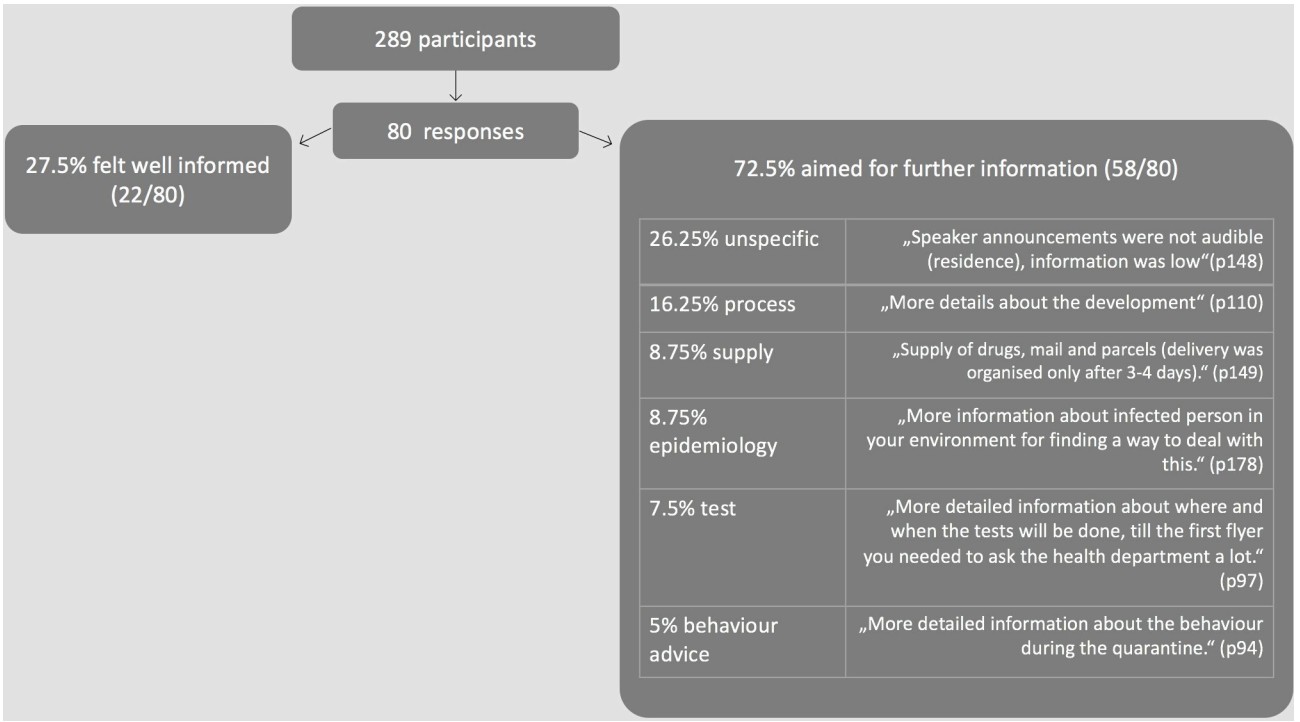

**Fig 2. Information needs of participants.**

As reasons for their own compliance, respondents stated that it was the required behaviour and that 'non-compliance-related occurrences led to a negative perception of the village. This made me furious' (participant, p. 90). They also understood the necessity. Respondents reported hearing about diseases and fatalities and experiencing fear ('I was very worried about civilisation as a whole. The media displayed horrific images from Italy and Spain (coffins, mass graves, etc.). I have frequently pondered whether and how the infection is altering the world and its inhabitants' (p. 97).

**Behavioural change.** Regarding a better understanding of infection control behaviour, participants reported that they persisted in avoiding places and people that they felt were suspicious, 'contaminated' or potentially dangerous, highlighting a grocery store, family and friends, crowds and medical facilities (Fig 3).

**Leadership.** Participants stressed the importance of local leadership and called on health officials to be present throughout the intervention ('Where was political leadership? Closed the village and done. That was not ok, and it has spread great uncertainty', p. 160). Local leadership plays a key role in risk communication and could ensure compliance ('Not even our mayor had a few words of encouragement for us. Sad!!!', p. 171). Another participant recommended open communication about the process and progress of solutions ('Open communication (. . . about) those regulations for supplies etc. (that) are still pending but are currently being developed.

**Table 2. Recommendations for communication during quarantine.**

| Category | Number of statements | Subcategory | Number of statements |
|---|---|---|---|
| Communication | 29 | Earlier announcement of the quarantine | 8 |
| | | Critical about the way the quarantine was announced | 14 |
| | | Need for using many different media outlets (digital, personal and analogue) | 7 |

**Table 3. Reasons given for quarantine non-compliance.**

| Section | Category | Example | Number of statements |
|---|---|---|---|
| **Intrinsic reasons** | Ignorance | Little to no intelligence (p. 167). | 5 |
| | Uncertainty | Because it came so suddenly, no one understood what was really happening (and how it will continue) (p. 145). | 6 |
| | Reasons given for non-compliance | Personal reasons were more important, it was lasting too long, not convinced of the situation (p. 104). | 9 |
| **Extrinsic reasons** | Social gatherings | Family celebrations, birthdays, etc. (p. 142). | 2 |
| | Emergencies | Emergencies, e.g. in the family (p. 151). | 1 |

(Communicate) that authorities, in particular at the district level, are partly overwhelmed because they had to act quickly and could never rehearse the case beforehand', p. 175).

Related to the transparency of the process and leadership, seven participants noted the relevance of *availability* (p. 107) and recommended *the explicit naming of a person in charge* (p. 180) or *telephone numbers and hotlines* (pp. 138, 158). Two participants recommended that an efficient information flow could have helped to reduce the feeling of isolation and stressed that physical isolation should not lead to social isolation.

## Recommendations for better quarantine management

Of the 289 participants, 154 (53%) provided recommendations for better public health management. We conducted a linear correlation with age but were unable to identify any age-dependent recommendations. The topics mentioned are categorised and summarised in Fig 4.

The most frequently mentioned recommendation was regarding coordination (116). Of these, 37 referred to control of the quarantine (27 recommended 'more control', ten demanded tougher punishments for quarantine violation), 26 were about supply (25 indicated the need to organise and secure supplies, one addressed medical care), 18 were on social restrictions

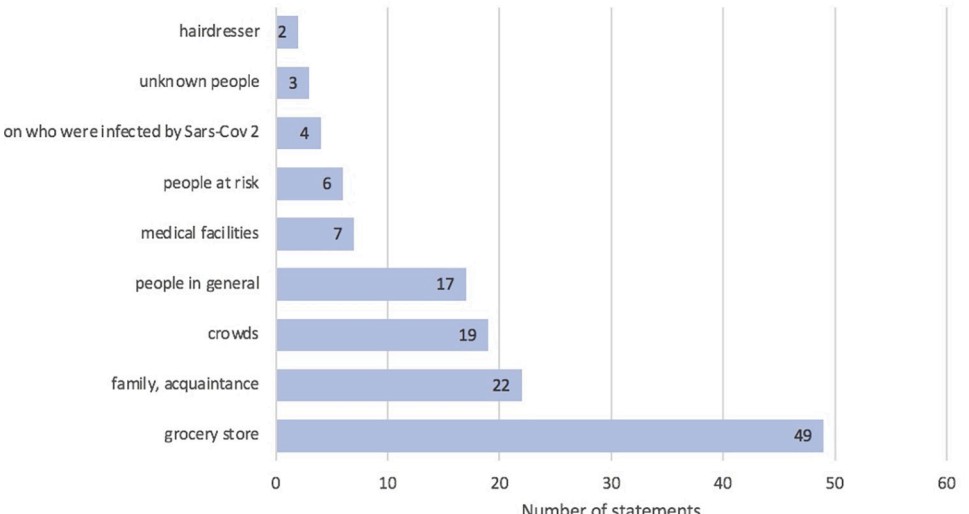

**Fig 3. Persistent avoidance of places and people.**

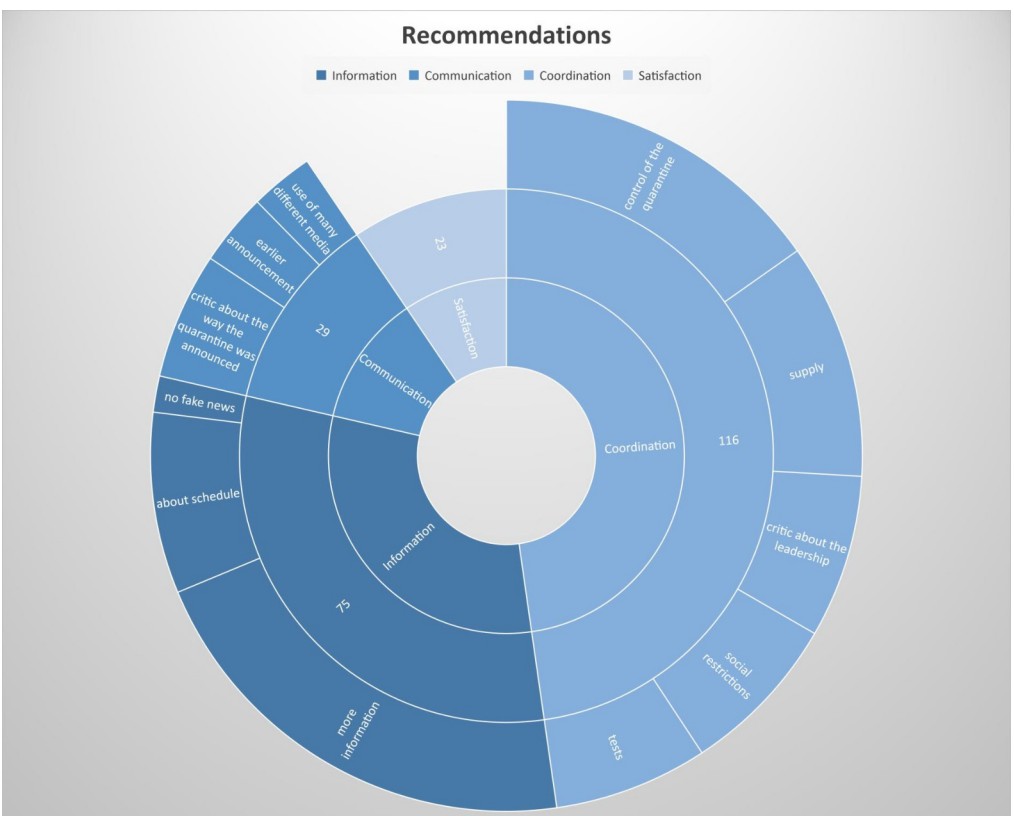

**Fig 4. Recommendations for successful quarantine management.**

(seven asked about expectations, five referred to distance keeping, three referred to face masks, two referred to a stricter quarantine and one recommended there should not be mass events), 18 criticised leadership and 17 were about tests (nine recommended organising more tests, five recommended better organising the tests and three recommended earlier testing).

Information was the second most frequently mentioned topic, with 75 recommendations. Of these, 51 asked for more information, 20 recommended being better informed about the schedule of the quarantine and four indicated that there should not be fake news.

Communication was addressed by 29 recommendations that included 14 critical observations about the way the quarantine was announced in Neustadt am Rennsteig. Eight participants recommended an earlier announcement and seven recommended using different media for communication.

Overall, 23 participants stated their satisfaction with the measures and the quarantine in Neustadt am Rennsteig, with recommendations such as 'do the same as in Neustadt' (p. 102).

## Discussion

Our survey touches upon relevant topics in the current discourse about risk communication during public health emergencies and the role of social media.

### Risk communication

**Information.** Respondents felt poorly informed. Based on their statements, there was a lack of fundamental and crucial information, which was primarily related to the effects of the

quarantine; for example: How do I purchase groceries? How do I get essential medication? How can I attend my medical appointments? For the authorities, the information initially centred on quarantine implementation. There was a mismatch between what authorities deemed important and what the population wanted to know. This disparity in information requirements is one of the fundamental challenges of communication, and risk communication in particular, and is one of the reasons why modern risk communication must first attend to and adapt to its target audience.

**Communication.** Only unidirectional communication channels (loudspeakers, flyers) were used, so those affected were unable to provide feedback or ask questions. These unidirectional means of communication also correlate with the compilation of information, which, as previously explained, corresponded more with the information requirements of the authorities than with those of the affected population. Participants emphasised that they would have preferred a more collaborative and participatory approach to risk communication and public health management, whereas public health stakeholders employed a command-and-control style. Successful risk communication and community involvement, however, require the emotional (empathy) and physical presence of those responsible and the participation of the impacted group in crisis management situations.

This disparity also sheds light on a novel aspect of the discussion regarding risk communication in health emergencies: In the past, the dichotomy between committed and commanding communication styles has received the most attention [20, 26]. However, this conflict highlights another significant issue: Risk communication has historically focused predominantly on individual health communication, in which individuals were advised on what they could do on an individual level, such as washing their hands or maintaining a safe distance.

In this study, we highlight the villagers' desire to be included, as an affected group, in decision-making and communication and their demand for an innovative leadership style. A group has an identity, established communication routines, formal and informal leaders and power in a functional social network. All these aspects could be used to improve risk communication. Yet in this unique communication circumstance, significant opportunities were missed.

In the literature, establishing or enhancing trust is cited as a benchmark of effective risk communication. Trust is the glue that holds communities and authorities together [9]. Beneficial communication strategies involve the community and utilise their local leaders to collaborate with authorities to gain trust [38]. Using local leaders ensures access to the affected communities [38].

Employing the internal structure of the group to gain access and to grow trust is a successful risk communication strategy. Both authorities and groups benefit from bi-directional communication. Authorities will be successful in their communication and will probably succeed with intended behaviour change; for groups, it affirms their internal group organisation and thus has a group stabilising function. To this end, this communication strategy promotes social coherence and enhances the resilience of the group [39].

For future risk communication in health emergencies, respondents unanimously and strongly recommended leveraging the group's strengths, established roles and communication media and routines.

**Coordination.** Despite inadequate information and communications, there was still, overall, considerate behaviour and basic agreement with the quarantine measure—right down to local patriotism (the Neustadt model: 'Do it like in Neustadt')(p. 102). Even if respondents complained about a lack of leadership and the absence of decision-makers, they reported that nearly everyone observed the quarantine. In addition, they adopted an acceptable degree of infection control behaviour.

People's initial avoidance of locations where there was a suspicion of infection has been described numerous times in the literature and can even be observed in animal populations [40]. Sites where infections have occurred are frequently perceived as menacing and stigmatised [5, 41]. The avoidance of Neustadt's (sole) supermarket was likely due to the presence of the outbreak's index case at the store's cash register. The avoidance of family and acquaintances differs from what is described in the literature. 'Misery loves company'—sharing misery in the community is a well-known strategy used by social organisations to successfully manage stress. The social distancing advice provided warnings about isolation during the quarantine and pandemic [30, 31, 42]. Throughout the pandemic, there were repeated demands to maintain a social distance and avoid other people, particularly crowds. It appears from the responses of the surveyed community that there is a distinction between personal/physical meetings and a close connection through social media. The WhatsApp group that people were connected to before the pandemic appeared to foster and maintain social coherence among this group. This social connection appears to be a potent resource for communication, group coherence and the emotional stability and resilience of the residents. The WhatsApp group seemed to compensate for the lack of physical personal interactions. This positive role of social media has not yet been prominently discussed. The mainstream discussion of social media use warns about the misinformation potential of social media.

As a result of this and other infectious disease epidemics, avoiding health facilities is a sensitive issue that has already been observed. During an outbreak, hospitals and medical practices are perceived as a source of infection and are avoided even after the epidemic has subsided [43–46]. However, not seeking healthcare for acute illnesses or preventive interventions poses a far greater risk. Risk communication should address the fact that healthcare avoidance is a significant health burden. To prevent this behaviour pattern from resulting in unfavourable collateral damage, balanced risk communication is necessary.

**Risk communication as a governance approach.** In our sample we noticed a strong connectedness of the villagers; they expressed the desire to be included in decision-making as a community and reflected on different communication and leadership styles. Risk communication, through this lens, has moved beyond being a communication technique for transmitting information from a sender (professionals) to a recipient (a target group) and has transformed into a governance concept.

Following this concept, risk communication at the individual level encompasses the information dimension and offers a bird's eye perspective on how the perspectives of authorities and communities can be better aligned.

Risk communication as a governance approach at the community level refers to communication in which the internal structure of groups can be better used for authorities to gain access and, eventually, to gain trust.

Risk communication at the coordination level refers to reflecting on the leadership of the leaders. Leadership and the existence of structural and systemic settings played an important role in the group's response.

Our participants were strongly connected using modern social media tools but complained about a lack of engagement and information from officials. Social media use for horizontal communication (communicating at the same level; e.g. among peers) is often perceived as contributing to misinformation. However, social media offers a strong platform for information, communication and even coordination activities. Our investigation demonstrated the positive role that social media tools played for the affected group. Rather than demonising social media activities, social media tools should become an integral part of public health risk communication.

## Strengths and limitations

Our study has several strengths and limitations. We investigated a small sample from a remote village in Thuringia. The sample size represents one-third of the entire population and is representative of this village. However, despite being small, we believed that we could observe important aspects relating to other observations and experiments in the field of modern risk communication approaches. These are relevant for future considerations to improve public health risk communication.

Participants who took part in our study may be more aware of risks than other residents, allowing for positivity and confirmation bias as special selection biases. This may even be more prevalent for the small group of participants who answered the qualitative part of the survey. The qualitative methodology allowed participants to voice their opinions and concerns.

Hotspots in remote areas were a key feature of this pandemic; investigating one of the first community quarantines offers important insights that could contribute to improving future public health management.

## Conclusion

Public health risk communication needs to employ and tap into research narratives and methodologies that detail how communities tick (community level) and how leadership can use modern communication tools and techniques (societal level) to better engage with their communities.

In our sample, participants wished for a more engaging and participatory risk communication and public health management approach. Yet, we observed a mismatch in both: information compilations and communication styles were mismatched and misaligned. The affected group had an identity (and boundaries), communication routines and tools, formal and informal leadership and power in its wider social network. These group features were not used by the authorities, resulting in difficulties accessing the group and gaining trust. In contrast to individual health communications, groups as organised sub-systems have connectivity and interaction opportunities. Public health risk communication should explore these opportunities more profoundly.

The role of social media is often seen in the context of misinformation. We could elaborate on the positive effect of social connection via the community WhatsApp group that offered necessary information and communication among peers and compensated for the lack of personal interactions during quarantine. These positive roles of social media tools should be explored and leveraged more widely in public health settings.

To this end, our study contributes to a novel understanding of risk communication that has evolved from a communication technique to a governance approach.

## Supporting information

**S1 File.**
(PDF)

## Acknowledgments

We wish to extend our appreciation to the population of Neustadt am Rennsteig, who participated in this study.

## The CoNAN study group

Technical University, Ilmenau, Germany: Thomas Hotz

Local cooperation partners, Neustadt am Rennsteig: Petra Enders, Renate Koch, Steffen Mai, Matthias Ullrich

Institute of Clinical Chemistry and Laboratory Diagnostics and Integrated Biobank, Jena University Hospital–Friedrich Schiller University, Jena, Germany: Cora Richert, Cornelius Eibner, Bettina Meinung, Kay Stötzer, Julia Köhler

Children's Hospital, Jena University Hospital–Friedrich Schiller University, Jena, Germany: Hans Cipowicz, Christine Pinkwart

Institute for Infectious Disease and Infection Control, Jena University Hospital–Friedrich Schiller University, Jena, Germany: Anita Hartung, Daniel Weiss, Lara Thieme, Gabi Hanf, Clara Schnizer, Jasmin Müller, Jennifer Kosenkow, Franziska Röstel

Institute of Immunology, Jena University Hospital–Friedrich Schiller University, Jena, Germany: Nico Andreas, Raphaela Marquardt

Institute of Medical Microbiology, Jena University Hospital–Friedrich Schiller University, Jena, Germany: Stefanie Deinhardt-Emmer, Sebastian Kuhn

## Author Contributions

**Conceptualization:** Wibke Wetzker, Petra Dickmann.

**Formal analysis:** Annika Licht, Wibke Wetzker, Petra Dickmann.

**Funding acquisition:** Petra Dickmann.

**Investigation:** Wibke Wetzker, Juliane Scholz, Petra Dickmann.

**Methodology:** Petra Dickmann.

**Supervision:** Petra Dickmann.

**Writing – original draft:** Annika Licht, Petra Dickmann.

**Writing – review & editing:** Wibke Wetzker, Juliane Scholz, André Scherag, Sebastian Weis, Mathias W. Pletz, Michael Bauer, Petra Dickmann.

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
