## [Decision Letter · Decision Letter 0]

27 Jun 2023

PONE-D-22-27696Public Health Risk Communication through the Lens of a Quarantined Community: Insights from a Coronavirus Hotspot in GermanyPLOS ONE

Dear Dr. Dickman,

Thank you for submitting your manuscript to PLOS ONE. After careful consideration, we feel that it has merit but does not fully meet PLOS ONE’s publication criteria as it currently stands. Therefore, we invite you to submit a revised version of the manuscript that addresses the points raised during the review process.

Please address the theoretical and methodological issues raised by the reviewers.

We look forward to receiving your revised manuscript.

Kind regards,

Rosemary Frey

Academic Editor

PLOS ONE

Journal Requirements:

"NO"

6. One of the noted authors is a group or consortium “the CoNAN study group”. In addition to naming the author group, please list the individual authors and affiliations within this group in the acknowledgments section of your manuscript. Please also indicate clearly a lead author for this group along with a contact email address.

7. Please ensure that you refer to Figures 1 an 2in your text as, if accepted, production will need this reference to link the reader to the figure.

Reviewers' comments:

Reviewer's Responses to Questions

**Comments to the Author**

1. Is the manuscript technically sound, and do the data support the conclusions?

Reviewer #1: Yes

Reviewer #2: Yes

2. Has the statistical analysis been performed appropriately and rigorously? 

Reviewer #1: I Don't Know

Reviewer #2: Yes

3. Have the authors made all data underlying the findings in their manuscript fully available?

Reviewer #1: Yes

Reviewer #2: Yes

4. Is the manuscript presented in an intelligible fashion and written in standard English?

Reviewer #1: Yes

Reviewer #2: Yes

5. Review Comments to the Author

Reviewer #1: An interesting study on public health risk communication from the perspective of a quarantined community.

The importance of social media needs to be further explained in the Introduction. Social media is a vital vehicle to provide timely health information and foster interactive communication by disseminating epidemic awareness, preventive measures, and emotional responses to the public (Budwani & Sun, 2020; Han & Xu, 2020; Liao et al., 2015; Roy et al., 2020). It's not clear how social media was used in risk communication in the study - it is said that inhabitants used social media to communicate with each other, so was there any impact of such communication?

It's not clear which risk communication model/theory/framework from the literature underpins this study. WHO is mentioned but there is no risk communication theory/model.

More practical implications of the study could be given in the Discussion, such as source credibility, which is essential for epidemic communication with the public. Research has shown that source credibility is positively related to the public’s trust of health information (e.g. Kostagiolas et al., 2014; Lu et al., 2021) which, in turn, affects the public’s response to the pandemic such as taking up preventive measures and sharing the information on social media (Lu et al., 2021; Vinck et al., 2019). Areas of further research can be given in the Discussion section too.

Reviewer #2: This is an interesting study that, in general, is clearly written and researched. The topic is obviously important, and will make a contribution to the related literatures as well as potentially influencing future communication strategies. There are a few clarifications needed before it is ready to publish.

38/ Quarantine is one of the most effective intervention / (should be 'interventions')

186-7/ The skewed age stratification of the studied case (nearly half in the village being over 60) introduces certain potential 'non-typical' aspects to this study, which raises the issue of the results also potentially being non-typical. So the description of Neustadt am Rennsteig as "a typical rural village" lacks credibility, as it presently stands, and the wording should therefore either be changed or some empirical justification given for it being "typical". (Surely, the German population as a whole doesn't have this skewed an age distribution, does it?)

205-212/ Yes the sample is, at a minimal level at least, sufficiently representative, but the question remains, 'representative' of what? Of this village. So any later broad generalizations of the results will tend to look weak and are questionably justified. Be careful not to extrapolate too much from this single case (village) to broader contexts, which may or may not be similar. These limitations in potential generalizability of the results need to be acknowledged as such in the limitations section.

Figure 1/ This is an unnecessary graphic and should be excluded, as all of this can & should be covered in the text of the methods section. Text mention is sufficient. (This is not a problem with the other graphics.)

245, 249 etc. If this is to go in a UK- or US-english journal, the ',' format for decimals should be changed to '.' format.

There is a potential problem with (likely) social desirability bias in responses to this survey. Self-report and opinion questions with regard to such a serious health crisis could be expected to reflect a greater-than-usual bias/distortion of this type. This limitation should be acknowledged in the limitations section.

Both the limitations section and the conclusion section are rather thin, as it stands. These need to be rewritten a bit more thoroughly to more completely explore implications of the research as well as its potential limitations.

6. PLOS authors have the option to publish the peer review history of their article (what does this mean?). If published, this will include your full peer review and any attached files.

Reviewer #1: No

Reviewer #2: No

---

## [Author Response · Author response to Decision Letter 0]

7 Aug 2023

Reviewers' comments:

Reviewer's Responses to Questions

Comments to the Author

1. Is the manuscript technically sound, and do the data support the conclusions?

Reviewer #1: Yes

Reviewer #2: Yes

2. Has the statistical analysis been performed appropriately and rigorously?

Reviewer #1: I Don't Know

Reviewer #2: Yes

3. Have the authors made all data underlying the findings in their manuscript fully available?

Reviewer #1: Yes

Reviewer #2: Yes

4. Is the manuscript presented in an intelligible fashion and written in standard English?

Reviewer #1: Yes

Reviewer #2: Yes

5. Review Comments to the Author

 

We, the authors, thank the reviewers for their thoughtful and constructive comments to our manuscript. We addressed each remark extensively and revised substantially most of the discussion part to reflect on the excellent comments of both reviewers. In addition, we used the opportunity to discuss the findings and the entire manuscript with some distance from the original time of submission. As requested, we re-submit two versions: one, with (extensive) track changes; two, a clean version for better readability. 

Again, we wish to extend our gratitude to the reviewers and their helpful remarks to improve our manuscript. 

Reviewer #1: An interesting study on public health risk communication from the perspective of a quarantined community.

1.1 The importance of social media needs to be further explained in the Introduction. Social media is a vital vehicle to provide timely health information and foster interactive communication by disseminating epidemic awareness, preventive measures, and emotional responses to the public (Budwani & Sun, 2020; Han & Xu, 2020; Liao et al., 2015; Roy et al., 2020). It's not clear how social media was used in risk communication in the study - it is said that inhabitants used social media to communicate with each other, so was there any impact of such communication?

We read, discussed and included the references above and broadened the aspect of social media in the manuscript. Especially in the introduction and the discussion part, social media use and its future role in public health risk communication is mentioned more in-depth. 

1.2 It's not clear which risk communication model/theory/framework from the literature underpins this study. WHO is mentioned but there is no risk communication theory/model.

We broadened the discussion about different risk communication approaches and models, but in favor of applicability for our public health audiences, we did not embark into a larger discussion on different risk communication theories. But we Included the participatory risk communication and community engagement (RCCE) approach published and applied by WHO and referenced it more precisely. We also introduced a pragmatic distinction between a traditional (sender – message – recipient model) and a modern risk communication governance approach. Our aim was to highlight important changes to risk communication practices, such as involving community leaders and to reframe social media use. 

While the reviewer is completely right in pointing to the accuracy of explaining different theories we felt that this manuscript has a more pragmatic, practice-oriented readership that we would lose if we discussed too much theory. 

1.3 More practical implications of the study could be given in the Discussion, such as source credibility, which is essential for epidemic communication with the public. Research has shown that source credibility is positively related to the public's trust of health information (e.g. Kostagiolas et al., 2014; Lu et al., 2021) which, in turn, affects the public's response to the pandemic such as taking up preventive measures and sharing the information on social media (Lu et al., 2021; Vinck et al., 2019). Areas of further research can be given in the Discussion section too.

The reviewer is completely right in highlighting the importance of credibility related to the public’s trust in health information, we thank him/her/them profoundly for the useful comment. We incorporated this remark and extended the discussion part.

Reviewer #2: This is an interesting study that, in general, is clearly written and researched. The topic is obviously important, and will make a contribution to the related literatures as well as potentially influencing future communication strategies. There are a few clarifications needed before it is ready to publish.

2.1 38/ Quarantine is one of the most effective intervention / (should be 'interventions')

Spelling corrected

2.2 186-7/ The skewed age stratification of the studied case (nearly half in the village being over 60) introduces certain potential 'non-typical' aspects to this study, which raises the issue of the results also potentially being non-typical. So the description of Neustadt am Rennsteig as "a typical rural village" lacks credibility, as it presently stands, and the wording should therefore either be changed or some empirical justification given for it being "typical". (Surely, the German population as a whole doesn't have this skewed an age distribution, does it?)

Clarified in the manuscript: rural villages have a different age distribution than cities in Germanies; with more older people. To this end, our study village is no exemption. We clarified this in the document (intro/methods and limitations). 

2.3 205-212/ Yes the sample is, at a minimal level at least, sufficiently representative, but the question remains, 'representative' of what? Of this village. So any later broad generalizations of the results will tend to look weak and are questionably justified. Be careful not to extrapolate too much from this single case (village) to broader contexts, which may or may not be similar. These limitations in potential generalizability of the results need to be acknowledged as such in the limitations section.

Very valid point. We adjusted the tone and included more reflection in the limitation section. 

2.4 Figure 1/ This is an unnecessary graphic and should be excluded, as all of this can & should be covered in the text of the methods section. Text mention is sufficient. (This is not a problem with the other graphics.)

Point well taken and moved to supplement.

2.5 245, 249 etc. If this is to go in a UK- or US-english journal, the ',' format for decimals should be changed to '.' format.

We proof read the entire document and took great care to write a consistent UK-English style. 

2.6 There is a potential problem with (likely) social desirability bias in responses to this survey. Self-report and opinion questions with regard to such a serious health crisis could be expected to reflect a greater-than-usual bias/distortion of this type. This limitation should be acknowledged in the limitations section.

We included this aspect in the limitations part. 

Both the limitations section and the conclusion section are rather thin, as it stands. These need to be rewritten a bit more thoroughly to more completely explore implications of the research as well as its potential limitations.

We extended both sections – discussion and limitations – significantly to reflect the helpful comments of the reviewers. Please see track changes for the changes or the clean document for the logical flow. We also rewrote the conclusion part.

---

## [Decision Letter · Decision Letter 1]

18 Sep 2023

Public health risk communication through the lens of a quarantined community: insights from a coronavirus hotspot in Germany

PONE-D-22-27696R1

Dear Dr., Dickmann

We’re pleased to inform you that your manuscript has been judged scientifically suitable for publication and will be formally accepted for publication once it meets all outstanding technical requirements.

Kind regards,

Rosemary Frey

Academic Editor

PLOS ONE

Additional Editor Comments (optional):

Reviewers' comments:

Reviewer's Responses to Questions

**Comments to the Author**

1. If the authors have adequately addressed your comments raised in a previous round of review and you feel that this manuscript is now acceptable for publication, you may indicate that here to bypass the “Comments to the Author” section, enter your conflict of interest statement in the “Confidential to Editor” section, and submit your "Accept" recommendation.

Reviewer #1: All comments have been addressed

Reviewer #2: All comments have been addressed

2. Is the manuscript technically sound, and do the data support the conclusions?

Reviewer #1: Yes

Reviewer #2: Yes

3. Has the statistical analysis been performed appropriately and rigorously? 

Reviewer #1: Yes

Reviewer #2: Yes

4. Have the authors made all data underlying the findings in their manuscript fully available?

Reviewer #1: Yes

Reviewer #2: Yes

5. Is the manuscript presented in an intelligible fashion and written in standard English?

Reviewer #1: Yes

Reviewer #2: Yes

6. Review Comments to the Author

Reviewer #1: Thank you for addressing the comments. I can see that some literature has been added and more practical implications have been given.

Reviewer #2: To my mind, the major necessary revisions in the manuscript itself have now been dealt with adequately, so (unless another reviewer still has reservations about the journal's technical or style requirements) it is strong enough for publication and will make an important contribution.

7. PLOS authors have the option to publish the peer review history of their article (what does this mean?). If published, this will include your full peer review and any attached files.

Reviewer #1: No

Reviewer #2: No
